# The Effect of Low-Doses of Caffeine and Taurine on Convulsive Seizure Parameters in Rats

**DOI:** 10.3390/bs10020043

**Published:** 2020-01-27

**Authors:** Mohamed Jailani, Mohamed Mubarak, Mariam Sarkhouh, Ahmed Al Mahrezi, Habib Abdulnabi, Mohamed Naiser, Husain Alaradi, Abdulaziz Alabbad, Maram Hassan, Amer Kamal

**Affiliations:** College of Medicine and Medical Sciences, Arabian Gulf University, Manama 1000, Bahrain; mohamed.a.jailani@gmail.com (M.J.); mohdkej@yahoo.com (M.M.); mmhsarkhouh@gmail.com (M.S.); almahreziahmed@yahoo.com (A.A.M.); drhabib1992@gmail.com (H.A.); mjnaiser@outlook.com (M.N.); hussainaradi4@gmail.com (H.A.); azeezalabbad@gmail.com (A.A.); maram.h.hassan97@gmail.com (M.H.)

**Keywords:** Caffeine, Taurine, Seizures, Pentylenetetrazole, seizure threshold, rats, adenosine

## Abstract

**Introduction**: Caffeine, an adenosine-receptor blocker, is believed to have neuronal excitatory effects, while Taurine, a mammalian amino acid, was shown to have neuroinhibitory effects. **Aim**: The aim of this study was to investigate the effects of acute and chronic administration of low doses of Caffeine and Taurine on the seizure threshold in rats. **Methods**: Six-week-old Sprague-Dawley male rats (n = 280) were divided randomly into five groups (control, acute caffeine intake, acute taurine intake, chronic caffeine intake and chronic taurine intake), with five subgroups per group according to five different doses of Pentylenetetrazole (PTZ) injections. Each subgroup consisted of eight rats. Data was entered and analyzed using Microsoft EXCEL and Addinsoft^TM^ XLSTAT (Version 2012.6.06; New York, NY, USA). *p*-value = 0.05 was regarded as statistically significant. **Results**: There was a significant decrease in the latency of PTZ-induced seizures with both acute (*p*-value < 0.05) and chronic (*p*-value < 0.01) Caffeine treatment groups. Chronic exposure to Caffeine exhibited an increase in the probability of seizures (*p*-value < 0.05). However, acute exposure to Caffeine did not show a significant impact on the probability of seizures. Neither acute nor chronic exposures to Taurine had an effect on the probability of seizures, nor on the latency of PTZ-induced seizures. **Discussion**: Our study found that acute as well as chronic exposure to low doses of Caffeine (50 and 80 mg/kg) reduces the threshold, and hence increases the likelihood for seizures since it favors a state of neuronal hyper excitability through blocking of all adenosine receptors. On the other hand, acute or chronic exposure to Taurine did not show a significant effect on the PTZ-induced seizures parameters.

## 1. Introduction

Abnormally excessive or synchronous neuronal activity is what causes the signs and symptoms of epileptic seizures [1]. Convulsive seizures constitutes the major type and may last from seconds to several minutes [2]. Seizure threshold is decreased by decreased sleep, excessive stress or medications with convulsant or threshold lowering substances, as well as caffeine [3].

Caffeine is metabolized by cytochrome P450 1A2 and its over-saturation due to excessive consumption or inhibition of the enzyme can lead to increased caffeine toxicity [4]. A moderate consumption of caffeine can induce CNS (central nervous system) and behavioral stimulations [5]. These effects are primarily related to its antagonism of adenosine receptors. Adenosine blocks the excitatory synaptic responses through modulation of neurotransmitter release and neuronal excitability by reducing potassium-mediated release of glutamate and aspartate [6].

Taurine (2-amino-ethanesulfonic acid), on the other hand is an amino acid found in in the excitable tissues in mammals [7,8]. It contains a sulfur group instead of a carboxyl group, which prevents it from being incorporated into proteins [7,9,10]. Several studies suggest that taurine’s nutritional effects include providing proper maintenance and functioning of skeletal muscles [11], removing fatty deposits in the liver, preventing liver diseases and reducing cirrhosis [12]. Taurine can readily cross the blood brain barrier and has a significant physiological role in the CNS [13]. It has been shown that taurine serves as a neuromodulator, neurotransmitter, neuroprotector against glutamate-induced neurotoxicity and a regulator of calcium binding proteins [14]. Taurine can play a role in reducing the spontaneous firing of neurons, hyperpolarizing the resting membrane potential, by increasing the membrane Cl- conductance [15,16].

Data demonstrated that caffeine exposure with low or high doses created a state of hyperexcitability through antagonism of the adenosine receptors and eventually favors seizure induction [17]. Other reports however, suggested that chronic caffeine exposure might increase neuronal adaptation and increase the resistance to seizure induction. Taurine on the other hand was shown to have inhibitory effects on seizure induction by some reports [15,16] and no effect in others, depending on the dose of the convulsant and the root of administration [18,19]

In order to systematically study the effects of these two chemicals, we constructed this experiment to include groups of animals that were exposed for short (acute) as well as long (chronic) periods of time to fixed doses of caffeine and taurine, and studied their effects on seizures induction by using different doses of pentylenetetrazole (PTZ).

## 2. Methods

### 2.1. Animals

The animals used were six-week-old Sprague-Dawley male rats (n = 280). They were divided randomly into five groups (control, acute caffeine intake, acute taurine intake, chronic caffeine intake and chronic taurine intake) with five subgroups per group according to five different doses of PTZ injections. Each subgroup consisted of eight rats. The animals were kept in a standard animal facility (12-hour light cycle with lights on at 7:00am) with food and water provided ad libitum.

### 2.2. Caffeine Administration

A total of 0.2 g/L of caffeine was dissolved in drinking water and was kept as the only source of water for two days in the acute intake group, and for one month in the chronic intake group [20]. Daily water consumption/rat/day was calculated and maximal caffeine daily dose was measured based on the animal weight. A mean caffeine intake of 14 mg/kg/day was estimated. This dosage corresponds to about 4 mg/kg/day in humans (which is about three regular cups of coffee) since the human metabolism of caffeine is different from in rats [21].

### 2.3. Taurine Administration

15 g/L of taurine was dissolved in the drinking water and was kept as the only source of water for two days in the acute intake group and for one month in the chronic intake group [22,23]. The doses of both caffeine and taurine were calculated by measuring the average daily consumption of water in seven days (bout 20 ml daily) and the body weight of the animals (about 300 g). Both caffeine (0.2 g) and taurine (15 g) were dissolved in one liter of water to provide assurance that the animals have the recommended dose of each substance.

### 2.4. Pentylenetetrazole Injections

Seizure threshold was tested by using epileptogenic Pentylenetetrazole (Sigma P6500) intraperitoneal injections with different doses. Testing the seizure threshold was done two days following caffeine/taurine consumption for the acute intake groups and one month following chronic caffeine/taurine consumption for the chronic groups. The different doses were 9, 18, 35, 50 and 80 mg/kg given as a single dose to the animal tested. The probability and latency of seizures were measured and compared between the groups.

### 2.5. Seizure Recognition

The induced seizures were identified blindly for the PTZ doses using the Racine Rating. Seizure latency was measured as the time lapse between the moment of PTZ-injection and the appearance of any of the following signs: mouth and facial movements, head nodding or forelimbs clonus with and without rearing and falling. The probability of seizure was estimated as the percentage of PTZ-induced seizure per group in relation to the total number of injected animals in that group. Seizure threshold refers to the dose of PTZ that could induce seizures in the animals.

## 3. Data Analysis

The data was expressed as mean ± SEM. The statistical significance was set at *p* = 0.05. All statistical analyses were carried out using Microsoft Excel and Addinsoft^TM^ XLSTAT (Version 2012.6.06; New York, NY, USA). Seizure probability was calculated the Chi-square test. Intergroup comparisons for seizure latency were made by using a two-tailed *t*-test.

## 4. Results

### 4.1. The Effect of Caffeine and Taurine Treatment on the Probability of PTZ-Induced Seizures

Seizures were induced with five different doses of Pentylenetetrazole (PTZ). The probability of a seizure; and subsequently the latency, were measured when the tested animal exhibited signs of a clonic seizure, which corresponded to stage 3 as per Racine stages. The difference in probability between control and groups treated with either taurine or caffeine showed no significant changes in the 9 mg/kg (*p* > 0.05), 18 mg/kg (*p* > 0.05) and 35 mg/kg (*p* > 0.05) doses.

#### 4.1.1. Acute Caffeine and Taurine Exposure

The data showed that no seizure was induced by very low STZ concentrations (9 and 18 mg/kg). However, higher doses (above 35 mg/kg) of STZ injections caused noticeable seizures with an increasing trend in terms of probability within the higher dose of 50 and 80 mg/kg (Figure 1). The acutely treated groups with caffeine and taurine did not show a statistically significant change in probability with 50 mg/kg STZ injections *(p > 0.05 Phi = 0.18).* There is however, an increased probability of seizure induction in the acute caffeine treated group when injected with 80 mg/kg of STZ. This figure shows that both caffeine and taurine did not affect the threshold of seizure induction. However, caffeine caused more intensive seizures than the other two groups with a high PTZ injection dose.

#### 4.1.2. Chronic Caffeine and Taurine Exposure

Similarly to for the acutely administered caffeine and taurine groups, low doses of STZ (9 and 18 mg/kg) did not cause any seizures. Among the chronic treatment groups, the probability of seizures in response to 50 mg/kg STZ injections was significantly higher in the caffeine treated group (*p* < 0.05, *Phi* = 0.55, *Chi-Square* = 12.52). In contrast, seizure probability was not significant in the taurine treated counterpart (*p* > 0.05, *Phi* = 0.11) when compared to the control groups (Figure 2). Rats injected with 80 mg/kg of PTZ exhibited an ascending trend in terms of seizure probability. Although there were probability higher values for convulsion in the chronic caffeine treated group, no statistically significant differences were recorded between the groups (*p* > 0.05, *Phi* = 0.35).

### 4.2. The Effect of Caffeine and Taurine Treatment on the Latency of PTZ-Induced Seizures

When the rats were administered 9, 18 and 35 mg/kg of PTZ, there was no significant difference in the latency period between the control group, and those that were under acute or chronic treatment of either taurine or caffeine (*p* > 0.05). These results were very similar to the probability of convulsions towards low doses of injected PTZ. In the group of rats exposed to 50 mg/kg of PTZ, the mean latency period in the control group was *404.63 ± 58.65* s. Despite showing a generally decreased latency time to seizure, the rats treated acutely with either caffeine or taurine exhibited no statistically significant changes in latency period (*p* > 0.05, *t* = 1.71; two tailed t-test) from the control group (Figure 3). On the other hand, chronic administration of caffeine resulted in a significantly shorter latency (*p* < 0.01, *t* = 4.37; two-tailed t-test) when compared with the control animals. However, chronic treatment with taurine (*p* > 0.05, *t =* 0.64; two tailed t-test) did not show any significant difference in latency periods from the control as well as the other acute treatment groups. As for the rats injected with 80 mg/kg of PTZ; the control exhibited a mean latency period of *281.93 ± 63.25* s. The groups acutely treated with caffeine showed a significant decrease in the latency period (*p* < 0.05, *t =* 3.26; two-tailed t-test) while those treated with taurine did not show any significant change from the control group (*p* > 0.05, *t* = −0.35; two-tailed t-test). Concerning the chronic groups; caffeine treated animals showed a significant reduction in the latency period (*p* < 0.01, *t* = 3.01; two-tailed t-test). The taurine treated animals, however, displayed no significant changes in the latency periods (*p* > 0.05, *t* = 0.07; two-tailed t-test). The data in this experiment shows that low PTZ injection did not show significant differences in term of seizures induction, thus not affecting the threshold of PTZ-induced convulsion. With higher doses of PTZ, it appeared that caffeine enhances and intensifying the PTZ-induced seizures, while taurine had no significant effect.

## 5. Discussion

In the present study, we found that caffeine consumption in a dose of 14 mg/kg, which corresponds to low caffeine consumption in humans (about three regular cups per day), was associated with alterations in seizure parameters. Both acute and chronic exposure to caffeine showed an increased probability of seizure induction and a decreased PTZ-induced seizure latency. However, opposite results were recorded by other groups in other experimental setups [24]. In addition, caffeine exposure had little effect on seizure latency and probability when the injected PTZ doses were very low. This may suggest that caffeine had no significant effect on seizure induction threshold. However, we showed that the induced seizures by higher doses of PTZ were aggravated in terms of seizure latency and probability by caffeine consumption. On the other hand, our study also showed that both acute and chronic forms of taurine consumption, even through an amino acid with neuroinhibitory properties, does not play a substantial protective role against seizures.

Caffeine has a psychostimulant effect on the central nervous system by antagonizing adenosine A_1_ and A_2A_ receptors that are found abundantly in the cerebral and cerebellar cortices, hippocampus and thalamic nuclei [25]. Adenosine A_1_ receptor inhibits neurotransmitter release by controlling the calcium currents [26]. In fact, its inhibitory effect is suggested to be stronger than that of all inhibitory neurotransmitters [27]. The basal nuclei which are the major motor inhibitory region in the brain, exerts its function using the adenosine _2A_ receptors by decreasing the affinity of dopamine to the D_2_ receptors [28]. Adenosine _2A_ receptors are also located in excitatory nerve terminals throughout the brain [29] where they control synaptic plasticity processes. [30]. Caffeine’s antagonism of adenosine receptors favors hippocampal hyperexcitability in the form of increased development of long-term potentiation. Higher doses of caffeine (100 mg/kg or more) provoke a hippocampal hyperexcitability which is similar to what is recorded during generalized seizures [31,32,33].

The physiological effects of caffeine, on a molecular level, confirms its role in favoring a state of arousal and neuronal hyperexcitability by antagonizing the inhibitory effects of adenosine at these receptors, and therefore precipitating a pro-epileptogenic state. Several other studies also suggest that caffeine plays a role in lowering seizure thresholds and can even directly induce seizures at high doses of 400mg/kg [34,35]. In fact, it has been also reported that caffeine consumption, in both low and high doses, has a negative influence on the protective effects of several anti-epileptic drugs such as gabapentin, topiramate, carbamazepine and valproate [36]. However, it was also mentioned that chronic consumption of caffeine could lead to adaptive physiological changes that cause reduced susceptibility to seizures, regardless of whether the seizure was induced by NMDA agonists or by GABA_A_ receptor antagonists such as PTZ [37,38].

In contrast to caffeine, taurine plays an inhibitory role in the central nervous system. It primarily exerts its effect through reducing glutamate-induced calcium influx and opening of chloride channels [36]. Studies suggest that pretreatment with taurine had anticonvulsant effects of pilocarpine-induced convulsions. This effect might be mediated by the reduction of acetylcholine esterase (AChE) and nitric oxide (NO) levels in the hippocampus [39,40]. Another study found that 43mg/kg subcutaneous taurine had a significant effect in decreasing the probability and increasing the latency of kianic acid induced tonic clonic seizures, in contrast to a non-significant change in the same parameters when the animals were supplemented with 0.05% taurine dissolved in water [18]. Although some studies [41,42] suggest that taurine has significant anticonvulsant properties, our findings did not support these results. A possible explanation as to why our trials of taurine supplementation failed to elicit significant anticonvulsant effects could be attributed to the route of taurine administration. The methods used by the previous studies included different and more invasive routes of administration. Our data agree with another study that 1.5% taurine supplementation in drinking water had no significant anticonvulsant effects [19].

In conclusion, our study results suggest that a low dose of caffeine is associated with a pro-epileptic effect in the form of an increased probability and decreased latency time to seizures. In contrast, taurine consumption seems to have no notable effect on the same seizure parameters. Considering these results, we believe that what remains to be addressed is the effect of higher doses of caffeine in acute, chronic and withdrawal exposures on similar and more specific seizure parameters such as the type of seizures occurring and brain electrical activity during and after the seizure. Our data indicate that caffeine consumption, especially in high doses, should be avoided in patients with epilepsy. However, there is discrepancy about this subject, since some reports support our results [43] while others demonstrated that caffeine does not appear to be a common seizure precipitant [44].

## Figures and Tables

**Figure 1 behavsci-10-00043-f001:**
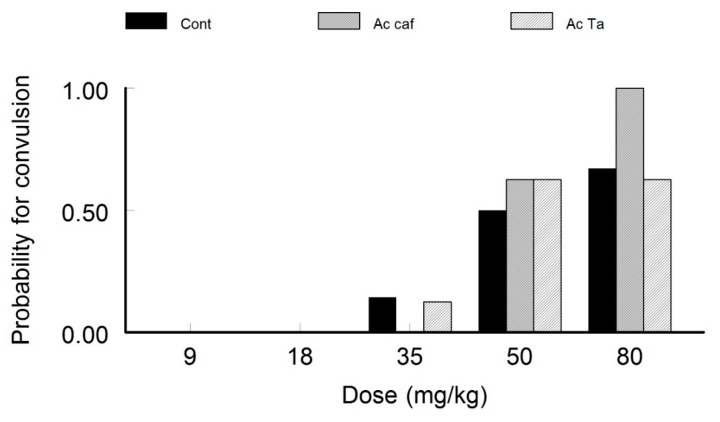
Probability of Convulsion in Acute Caffeine and Taurine Exposure. Low doses of the convulsant pentelynetetrazole (PTZ) injections did not cause any seizures. With higher doses of PTZ-injection, the acutely exposed animals to caffeine showed an increased probability of seizures compared to the other groups. Taurine did not protect the animals from seizure induction.

**Figure 2 behavsci-10-00043-f002:**
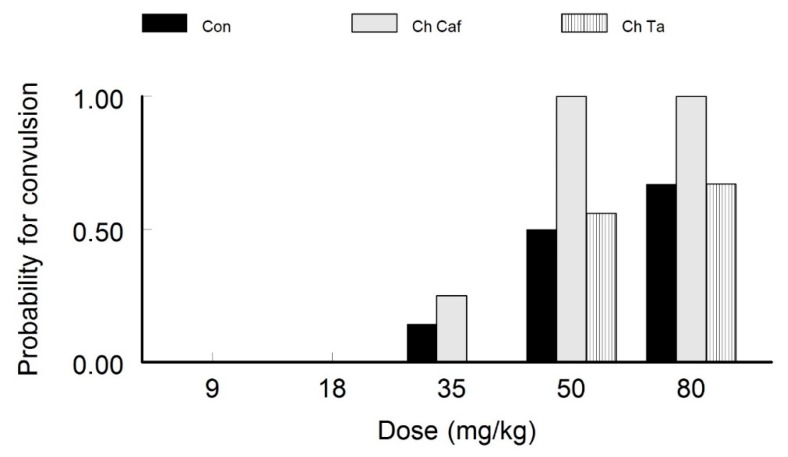
Probability of Convulsion in Chronic Caffeine and Taurine Exposure. Low doses of PTZ did not cause seizure in all the groups. However, with high PTZ-injection doses the group exposed to chronic caffeine consumption demonstrated increased susceptibility and showed a significantly increased probability of PTZ-induced convulsion.

**Figure 3 behavsci-10-00043-f003:**
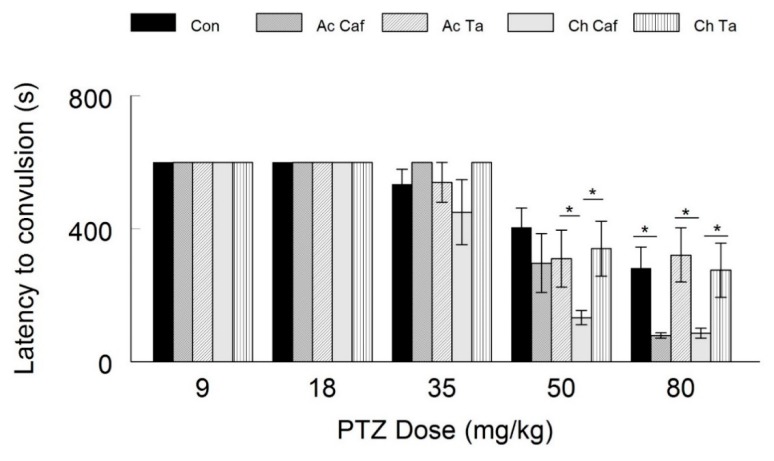
Latency Period to Convulsions. No significant difference was calculated between the groups in the latency to convulsion when the injected PTZ was of low doses. Higher doses of injected PTZ caused convulsions with significant low latency, and thus more intense seizures, in the acute and chronic caffeine consumption groups when compared to the other groups. Taurine did not exert any protecting effect against the convulsing effect of the injected PTZ.

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
