# Peer review of "The Effect of Low-Doses of Caffeine and Taurine on Convulsive Seizure Parameters in Rats"

_behavsci, 2020, doi:10.3390/bs10020043_

Round 1

Reviewer 1 Report

In the manuscript, the authors investigated the effects of caffeine and taurine on the susceptibility to seizures induced by PTZ. They found that caffeine caused increased seizure probability an decreased seizure latency in PTZ-induced seizures, while taurine did not have significant effect.

Here are my comments.

1.The manuscript lacks novelty. Similar experiments have been conducted and similar results have been found in other models.

The results are not convincing. The sample number (8 rats for each group) is too small for the analysis of PTZ-induced seizure which usually has a big variability in seizure parameters. to In methods, it is mentioned that “ 1g/Kg/day taurine was dissolved in drinking water”. What does it mean? Figure 3 is not presented properly. It is hard to tell what those asterisks indicate. There are too many grammatic errors in the manuscript.

Author Response

Good afternoon.

We thank very much the effort of the reviewer and all his/her critics and remarks.We believe the intention was to make our manuscript of a level acceptable for publication.

In the new version:

The whole paper was revised by an English expert and all needed editions were made.

The Introduction was re-written as requested.

The methods section was re-written. Concerning the doses of caffeine and taurine: The doses and the way of their estimation were carefully explained. Please see the method section for this.

About the number of animals used in the experiment: May be it was not very clear in the manuscript. There were seven groups of animals. Each group had actually 5 subgroup (we used five different concentrations of STZ injections), and one of these subgroups contained 8 animals. So the total number of animals / group was 8 *5= 40 (not 8). This makes the total number of rats in this experiment (40 * 7= 280 rats).

The results were re written to clarify things that were not, and description of data and statistics are now more clear.

The Discussion section was edited in a more clear way.

Concerning that this subject was visited and searched before by other groups and data were published in this area in other models, and that our result are just like others:

1- We mentioned that opposite results in some of the data were reported by others (for example, ref 20). However, research is Re- Search, and experimental designs with other type of animals, or other protocols are always beneficial to confirm or show different results. We used here two widely used substances and in different doses and investigated the effect of their use on convulsion threshold. One is known as stimulant and other was inhibitory or calming. Several doses were used. This set up as long as we know was not used before, or may be in different set ups.

2- We showed that taurine did not show protective effect on seizures, while other reports did.

3- Some reports showed that caffeine by itself can induce convulsion. Our results indicated that caffeine does not cause convulsion but may aggravate the STZ-induced convulsion.

4- Some reported that chronic caffeine consumption may reduce the susceptibility to seizure due to adaptive changes, but we did not notice that.

5- We did not find that taurine has anticonvulsant effect, while others did. Our explanation was that the root of administration was different.

Reviewer 2 Report

The authors describe a series of experiments that examine the effects of caffeine and taurine consumption in rats on their susceptibility to seizures triggered by pentylenetetrazole. They find that in doses of caffeine in both the acute and chronic conditions increased the likelihood of seizures, as well as reduced the latent time prior to seizure onset. Conversely, taurine had no significant effects on pentylenetetrazole-induced seizures.

The experiment is well-designed and the authors provide a plausible mechanism by which caffeine lowers the threshold for seizure onset by increasing the excitability of the central nervous system.

My only feedback is that certain English expressions should be corrected. For example:

"worthy to mention" should be "worth mentioning"

and

"Neither acute nor chronic exposures" should be "Neither acute nor chronic exposure"

Otherwise, I recommend that this manuscript be accepted in its present form.

Author Response

We thank very much the reviewer with his positive encouraging remarks. We did all the suggested point he raised.

We hope the manuscript in its new edited form is suitable for publication.

Round 2

Reviewer 1 Report

The manuscript has improved significantly after revision. Please correct "STZ" to "PTZ" in the methods section.

Author Response

Mistakes in the spelling were corrected as good as possible.
